# Advantages in Wound Healing Process in Female Mice Require Upregulation A_2A_-Mediated Angiogenesis under the Stimulation of 17β-Estradiol

**DOI:** 10.3390/ijms21197145

**Published:** 2020-09-28

**Authors:** Felipe Troncoso, Kurt Herlitz, Jesenia Acurio, Claudio Aguayo, Katherine Guevara, Fidel Ovidio Castro, Alejandro S. Godoy, Sebastian San Martin, Carlos Escudero

**Affiliations:** 1Vascular Physiology Laboratory, Department of Basic Sciences, Universidad del Bío-Bío, 3780000 Chillán, Chile; fetronc@gmail.com (F.T.); kherlitz@gmail.com (K.H.); jeseniacurio2@gmail.com (J.A.); kattyguevarae4@gmail.com (K.G.); 2Programa de Doctorado en Ciencias Veterinarias, Universidad de Concepción, 3780000 Chillán, Chile; 3Group of Research and Innovation in Vascular Health (GRIVAS Health), 3780000 Chillán, Chile; caguayo@udec.cl (C.A.); alejandro.godoy@uss.cl (A.S.G.); sebastian.sanmartin@uv.cl (S.S.M.); 4Department of Clinical Biochemistry and Immunology, Faculty of Pharmacy, University of Concepción, 4030000 Concepción, Chile; 5Department of Animal Science, Faculty of Veterinary Sciences, Universidad de Concepcion, 3780000 Chillán, Chile; fidcastro@udec.cl; 6Department of Urology, Roswell Park Comprehensive Cancer Center, Buffalo, NY 14263, USA; 7Centro de Biología Celular y Biomedicina (CEBICEM), Universidad San Sebastián, 8320000 Santiago, Chile; 8Biomedical Research Centre, School of Medicine, Universidad de Valparaíso, 2340000 Valparaíso, Chile

**Keywords:** adenosine, A_2A_ receptor, angiogenesis, wound healing

## Abstract

Estrogenic steroids and adenosine A_2A_ receptors promote the wound healing and angiogenesis processes. However, so far, it is unclear whether estrogen may regulate the expression and pro-angiogenic activity of A_2A_ receptors. Using in vivo analyses, we showed that female wild type (WT) mice have a more rapid wound healing process than female or male A_2A_-deficient mice (A_2A_KO) mice. We also found that pulmonary endothelial cells (mPEC) isolated from female WT mice showed higher expression of A_2A_ receptor than mPEC from male WT mice. mPEC from female WT mice were more sensitive to A_2A_-mediated pro-angiogenic response, suggesting an ER and A_2A_ crosstalk, which was confirmed using cells isolated from A_2A_KO. In those female cells, 17β-estradiol potentiated A_2A_-mediated cell proliferation, an effect that was inhibited by selective antagonists of estrogen receptors (ER), ERα, and ERβ. Therefore, estrogen regulates the expression and/or pro-angiogenic activity of A_2A_ adenosine receptors, likely involving activation of ERα and ERβ receptors. Sexual dimorphism in wound healing observed in the A_2A_KO mice process reinforces the functional crosstalk between ER and A_2A_ receptors.

## 1. Introduction

A substantial body of evidence describes contrasting influences of androgenic and estrogenic sex steroids on the healing of acute skin wounds, in which the former inhibits whereas the latter accelerates recovery [1]. However, gender differences in wound repair parameters would not intuitively be expected [2,3]. Since healing is a complex mechanism involving several processes, such as inflammation, coagulation, and angiogenesis, among others, it is feasible that female mice may have certain advantages in some of those processes.

Angiogenesis is a tightly regulated process that is rare under normal conditions in adults, except in the female sexual tract where it occurs periodically as a result of fluctuations in sex hormone levels [4]. Among many cell actors in the angiogenesis process, endothelial cells (ECs) have an essential role for new vessel formation via an increase in cell proliferation, migration, and tube formation capacity [5].

An increasing body of evidence indicates sexual dimorphism in endothelial cell function and therefore in the angiogenesis process [6,7]. Mechanistically, estrogen (E2) directly modulates angiogenesis [8], mostly through the activation of the classic estrogen receptors (ERs) alpha (ERα) [9] and beta (ERβ) [10] in endothelial cells. For instance, 17β-estradiol enhances proliferation, migration, and tube formation of endothelial cells, an effect likely involving the activation of the vascular endothelial growth factor (VEGF) and ERα pathways [9].

On the other hand, adenosine is also an endogenous modulator of angiogenesis in a cell-dependent manner via its four adenosine receptors (AR), named A_1_, A_2A_, A_2B_, and A_3_ [11,12]. In particular, A_2A_ (and A_2B_) receptors increase endothelial cell proliferation and migration in humans [13,14,15], pigs [16], and rats [17]. Moreover, A_2A_ also increases the functional expression of VEGF in endothelial cells [18,19,20]. Accordingly, the key role of the A_2A_ receptor in angiogenesis has been confirmed using A_2A_-deficient mice (A_2A_KO), which showed reduced blood vessel and extracellular matrix formation [21] and a delayed in vivo wound healing process [22] compared to wild type mice (WT).

Both estrogen (i.e., 17β-estradiol) via ER and adenosine via A_2A_ activation promote angiogenic behavior of endothelial cells. However, there is no information regarding potential crosstalk between these two main regulators. Interestingly, ovariectomy causes a dramatic reduction of the mRNA levels of A_2A_ receptors [23], suggesting that expression of the adenosine A_2A_ receptor may be regulated by estrogen. Indeed, 17β-estradiol upregulates the mRNA levels of A_2A_ in a dose-dependent manner in the human cancer cell line MCF-7 [24]. Complementary to this in vitro evidence, it has been shown that mice lacking A_2A_ receptors exhibited sex dimorphism in many physiological processes including heart rate, temperature, locomotion activities, oxygen demand [25], and maturation of the microglia in the brain [26], in which female A_2A_KO mice were less negatively affected than male A_2A_KO.

Using an in vivo wound healing assay and a primary culture of pulmonary endothelial cells isolated from female and male wild type (WT) and A_2A_KO mice, we aimed to investigate whether female mice exhibited advantages in the wound healing process and angiogenesis. Additionally, we wanted to investigate whether 17β-estradiol regulates the expression and pro-angiogenic function of the A_2A_ receptor and to elucidate whether ER and VEGF are involved in this regulation. Our a priori hypothesis indicates that 17β-estradiol potentiates the A_2A_-mediated angiogenesis in female derived endothelial cells that involve ER and VEGF activation and promote an advantage in the healing processes compared to male mice.

## 2. Results

### 2.1. In Vivo Wound Healing Assay in Female and Male Wild Type and A_2A_-Deficient Mice

Wound healing requires the presence of an A_2A_ receptor (Figure 1A). No significant differences were observed between female and male WT or female and male A_2A_KO mice, although female WT and female A_2A_KO mice tend to have a faster wound healing process compared to their respective male counterparts.

Faster wound healing was found in female WT mice than in female A_2A_KO mice at days 8 and 10 of the analysis. This apparent advantage in female WT mice was present even early (from day 4 to day 10) compared to male A_2A_KO mice (Figure 1B; *p* < 0.05 in all comparisons). Nevertheless, male WT showed higher healing capacity compared to male or female A_2A_KO only at day 10. This means that male A_2A_KO mice exhibited the lowest wound healing capacity.

We also analyzed skin blood perfusion and quantification of blood vessels at day four after injury in the dermis of the wounded area using Laser Doppler and histological analysis, respectively. A_2A_KO mice tended to have reduced blood perfusion (Appendix A; *p* = 0.06) associated with a reduced number of blood vessels in the dermis of the wounded area compared to WT mice (Appendix A). Sex dimorphism was observed only in WT mice and solely in the blood perfusion studies (Appendix A) but not in the blood vessel count (Appendix A) in which male WT exhibited higher perfusion (i.e., redness due to the inflammatory healing process) than female WT (*p* < 0.05).

### 2.2. Characterization of Mice Lung Endothelial Cells (mPEC)

Since previous reports showed a more pro-angiogenic behavior in female than male endothelial cells [7,27], we isolated female and male pulmonary endothelial cells (mPEC) from WT mice and A_2A_KO mice (Figure 2). Primary cultures enriched in endothelial cells were successfully established as demonstrated by the detection of the endothelial markers KDR and CD34 using Western blot analysis (Figure 2A,B). Levels of KDR increased by 184% and 194% after magnetic immunoselection using CD31 Dynabeads (CD31^+^ cell) in cells isolated from WT or A_2A_KO mice, respectively. Likewise, CD34 expression augmented 299% and 341% in CD31^+^ cells. No statistically significant differences were found in both endothelial markers when WT and A_2A_KO groups were compared, neither previous nor after immunoselection. Cell gender was confirmed by PCR amplification of the *Jarid* gene, while A_2A_KO origin of the cells was confirmed by the presence of the neomycin cassette (Figure 2C).

Furthermore, cells were used for in vitro angiogenic capacity (Figure 2D,E). CD31^+^ cells isolated from female and male WT and A_2A_KO mice presented a similar capacity to form tubular structures as early as 4 h after seeding them on Matrigel. The cells exhibited growth inhibition by cell-to-cell contact (data not shown).

Analysis of the transcript levels of the adenosine receptors A_1_, A_2B_, and A_3_ in the primary culture of mPEC from A_2A_KO mice revealed no statistically significant differences in any of the receptors compared to WT counterparts. However, when the analysis was carried out considering sex, male WT exhibited the highest expression of A_3_ receptor mRNA (*p* < 0.05) vs. female WT (Figure 2F).

### 2.3. Sex Dimorphism in A_2A_ Adenosine Receptors Expression

Mice pulmonary endothelial cells from female and male WT mice were used for the analysis of the A_2A_ expression. Higher mRNA (Figure 3A) and protein (Figure 3B) levels of A_2A_ were observed in mPEC isolated from female WT mice when compared to male WT mice.

Conversely, 17β-estradiol (10^−7^ M, 24 h) reduced the A_2A_ protein levels in both mPEC isolated from female WT mice (Appendix A) and in human umbilical vein endothelial cells (HUVEC) isolated from female babies (Appendix A).

### 2.4. 17β-Estradiol Enhanced Both A_2A_-Independent and A_2A_-Dependent Cell Proliferation in Female WT Mice

Cells from male WT mice showed reduced proliferation in the presence of 17β-estradiol (Figure 3C). Conversely, 17β-estradiol (24 h) enhanced cell proliferation in cells derived from female WT mice in a dose-dependent manner (Figure 3D). This was an effect that was similar to A_2A_-mediated cell proliferation, since 17β-estradiol and CGS-21680 showed similar LogEC_50_ (−7.40 M and −7.17 M, respectively) (Table 1). Importantly, the co-incubation of 17β-estradiol + CGS-21680 had a synergic effect, showing a shift of at least one order of magnitude in the log EC_50_ (Figure 3D and Table 1).

Similar to the use of the A_2A_ selective agonist CGS-21680 in cells from female WT mice, the co-incubation of 17β-estradiol (10^−7^ M) alongside the non-selective adenosine receptor agonist NECA (10^−5^ M, 24 h) showed a higher response than NECA alone. This was prevented in the presence of the A_2A_ antagonist, ZM-241385 (Figure 3E). ZM-241385 alone did not affect cell proliferation.

### 2.5. 17β-Estradiol Enhanced A_2A_-Dependent Angiogenesis in Female WT Mice

We further explored the potential relationship between estrogen and A_2A_ receptor on endothelial cell proliferation using mPEC isolated from female and male WT and A_2A_KO mice. At basal conditions, no significant differences in cell proliferation were found in female or male mPEC derived from WT or A_2A_KO mice (Figure 4A). However, in response to NECA (Figure 4B) or CGS-21680 (Figure 4C), female mPEC from WT mice showed a higher sensitivity to both agonists. This was observed as a left shift of at least one order of magnitude in the dose–response curve when compared to cells derived from male WT (Table 2).

To confirm the participation of the A_2A_ receptor in the sex dimorphism observed in cell proliferation, we used the selective A_2A_ antagonist ZM-241385 (Figure 4D) or analyzed cell proliferation in mPEC derived from A_2A_KO mice (Figure 4E). ZM-241385 prevented the augmented NECA-induced proliferative response observed in mPEC isolated from female WT mice. While sex dimorphism observed in NECA-induced cell proliferation was observed in WT mice, it was absent in A_2A_KO mice.

Compatible with our last results, mPEC from female WT mice exhibited higher CGS-21680-induced cell migration (Figure 5A) and tube formation (i.e., angiogenesis) (Figure 5C) than cells from male WT mice. However, again, this sex dimorphism observed in WT was absent in cells isolated from A_2A_KO mice (Figure 5B,D).

Furthermore, NECA and CGS-21680 enhanced VEGF protein levels (Figure 6) in cells from female WT, an effect that was absent in cells from female A_2A_KO mice. Indeed, VEGF levels were severely diminished in female A_2A_KO mice compared to female WT in all experimental conditions.

#### Estrogen-A_2A_ Synergic Effect Involves ERs

Finally, we further analyzed whether the synergic effect of 17β-estradiol + CGS-21680 observed in cell proliferation found in female WT was mediated by estrogen receptors. After confirming that female WT mice express both ERα and ERβ (Figure 7A,B), we found that the synergic effect observed on cell proliferation using CGS-21680 + 17β-estradiol was blocked when cells were co-incubated with the selective antagonists for both ERα and ERβ (Figure 7C).

## 3. Discussion

The adenosine A_2A_ receptor has a well-described pro-angiogenic role, but it is unknown whether its expression and activation are influenced by estrogen in endothelial cells. We showed that female mPEC has a higher expression of A_2A_ receptors. 17β-estradiol enhanced the endothelial cell proliferation induced by A_2A_-stimulation, an effect more likely to be associated with the activation of both ERα and ERβ receptors. Additionally, we found that the pro-angiogenic behavior mediated by stimulation of A_2A_ showed a sex dimorphism in mPEC isolated from WT mice, with female cells being more sensitive to an A_2A_-mediated response. This female advantage was absent in cells isolated from A_2A_KO mice. In vivo confirmation of sex dimorphism showed that A_2A_-deficient mice exhibited a delayed healing process and fewer blood vessels in the skin. According to our in vitro experiments, female WT mice have a more rapid wound healing process than A_2A_KO mice, suggesting a crosstalk between estrogen and A_2A_ receptors. However, the underlying mechanisms of this potential ER-mediated regulation of A_2A_ receptor expression and function was poorly understood.

Sex steroids regulate the healing process of acute skin wounds [1]. Sex dimorphism is present in endothelial function, as demonstrated by the capacity of synthesis of nitric oxide [7], a key molecule involved in vascular tone regulation as well as angiogenesis [6]. It is unclear whether this apparent advantage present in females also involves the wound healing process. In particular, the dorsal incisional wound healing process—generated in a similar experimental setting in this manuscript—showed no difference between C57BL/6 male and female mice after 50 days post-injury [2]. In this last report, female mice tended to have more rapid wound healing, especially in the time-lapse of up to 20 days. Using another model of tissue recovery and angiogenesis, such as the hind limb blood flow recovery after femoral artery ligation, it was found that female C57BL/6 mice had impaired hind limb use on day seven after the artery ligation compared to their male counterpart [3]. Contrary to our findings, this last piece of evidence suggests that female mice might have a reduced healing process in comparison to male mice. Since, healing is a complex mechanism involving several other processes, including activation of inflammation, coagulation, angiogenesis, and matrix recovery, among others, it is feasible that female mice may have certain advantages in some of those processes (i.e., angiogenesis), but not in all of them. In line with this observation, specific differences such as higher macrophage infiltration [1] or collagen (type I and III) synthesis [28] in the healing area were found in females when compared to male mice.

Under this complexity, sex dimorphism in some cardiovascular functions [25] or the brain maturation process [26] has already been described in A_2A_KO mice. We contribute to those findings by indicating that sex dimorphism may also be present during the wound healing process and tissue perfusion. Although no differences were found in wound healing between female and male A_2A_KO mice, male, but not female, A_2A_KO mice have a more delayed rate of healing compared to female WT. Our results disagree with previous reports in the same strain of A_2A_KO mice that showed no differences in the wound healing of those mice compared to their WT counterpart [22], although the authors did not analyze sex dimorphism. Despite that, the authors also reported defects in the formation of granulation tissue and a reduction in the number of Factor VII-positive endothelial cells at days three or six after dermal excisional wounds in A_2A_KO mice. Contrary to WT, A_2A_KO mice did not develop bleomycin-induced dermal fibrosis [21]. Then, it was confirmed that an A_2A_ receptor was required to synthesize the dermal extracellular matrix. Therefore, A_2A_ appears to control the formation of two key components of the tissue microenvironment, such as the extracellular matrix and blood vessels (i.e., angiogenesis). However, the potential impact of the lack of A_2A_ in the healing process would require confirmation in a larger number of samples and more time-extended analyses.

We found significantly fewer blood vessels in the A_2A_KO mice compared to WT mice on day four after injury, which may be related to less blood perfusion. In support of this finding, a reduced number of endothelial cells (i.e., blood vessels) in the walls of air punches of A_2A_KO mice were shown [22]. We extended the current knowledge by showing that blood perfusion at day four after injury showed sex dimorphism in WT mice, similar to what was found previously in the hind limb perfusion after femoral artery ligation [3]. In particular, high blood perfusion was found at day four after injury in male WT mice compared to their female counterparts, which may correspond to the initial phase of healing processes that are characterized by excessive angiogenesis accompanied by an increase in blood flow [29], while in female mice, this healing process and angiogenesis may be accelerated, and then reduced blood perfusion may constitute indirect evidence of this accelerated process. Interestingly, this sexual dimorphism observed in blood perfusion in the wounded area was absent in A_2A_KO mice, confirming that A_2A_ was required for adequate blood vessel formation and action. How estrogen or sex hormones contribute to control blood perfusion during the healing process in the absence of A_2A_ receptors is still unclear.

Our results suggest that sex hormones, including estrogen, may regulate the expression and activity of A_2A_ receptors in endothelial cells. As far as we know, there are no previous reports on this topic. Previous evidence using a human cancer cell line (MCF-7) showed that 17β-estradiol upregulates the mRNA levels of A_2A_ in a dose-dependent manner, an effect that was inhibited by the ER antagonist ICI182780 [24]. Additionally, whole-brain extracts of female rats exposed to ovariectomy revealed subtype-specific repression of adenosine receptors three months after surgery, with preferential downregulation of A_2A_ (4.3 fold), A_3_ (2.3 fold), and A_1_ (2.1 fold), but not A_2B_, receptors [23]. Instead, we showed that 17β-estradiol significantly reduced the total protein amount of A_2A_ in mice or human female endothelial cells.

Nevertheless, we also found that endothelial cells derived from female WT mice had higher mRNA levels of A_2A_ than cells from male WT mice, which was associated with 17β-estradiol upregulation of A_2A_-mediated cell proliferation. This observation may be interpreted as a counterintuitive finding considering the downregulation of the total protein amount of A_2A_ induced by 17β-estradiol in female endothelial cells. However, these findings may also suggest a regulatory loop between ERs and A_2A_ receptors in female endothelial cells, which might involve both transcriptional and translational regulation. In addition, since functional potentiation between 17β-estradiol + CGS-21680 (A_2A_ selective agonist) was found in our results, we encourage future studies focused on the 17β-estradiol regulation of intracellular traffic of the A_2A_ receptor, its location on the cell membrane, or A_2A_-derived intracellular pathways.

Our results also suggest that estrogen-mediated upregulation of A_2A_ receptor activity could be mediated by either ERα or ERβ, since the synergic effect of 17β-estradiol + CGS-21680 in mPEC proliferation was blocked with the respective ER antagonists. In this regard, using MCF-7 breast cancer cells, A_2A_ adenosine receptor crosstalk with ERα has been described in the regulation of the expression of progesterone receptor (PR), a well-described target of ERα. Thus, the expression of PR induced by CGS-21680 was inhibited with the ERα antagonist, ICI 182,780 [30]. In accordance with our results, 17β-estradiol and CGS-21680 had a similar proliferative effect on MCF-7 cells, which was interpreted as a part of the crosstalk between ERα and A_2A_ receptors [30]. As far as we know, no other reports have suggested a direct interaction between ER and A_2A_ receptors.

To confirm the relevance of A_2A_ in the pronounced pro-angiogenic behavior of female endothelial cells compared to male cells, we found that female mPEC derived from A_2A_KO lost the NECA or CGS-21680 mediated pro-angiogenic advantages (demonstrated in the form of higher cell proliferation/migration and tube formation capacity) present in female cells isolated from WT. These results support the hypothesis of a crosstalk between A_2A_ and ERs, although the underlying mechanism is still unknown. Since previous evidence described that estrogen [9] or A_2A_ receptor [18,19,20] independently upregulated VEGF, we decided to analyze this factor in our experimental setting. Thus, mPEC from female WT mice showed a threefold increase in VEGF levels compared with mPEC from female A_2A_KO mice. We could speculate that, in our experimental setting, the major regulator of VEGF was the A_2A_ receptor rather than estrogen. Therefore, experiments in ovariectomized or ER-deficient mice must be conducted. Nevertheless, both CGS-21680 and NECA upregulated VEGF protein levels in cells from female WT mice, again suggesting a potential crosstalk between ER and A_2A_ receptors.

Sex dimorphism was also found in A_3_ receptor expression in mPEC from WT mice, a phenomenon that was not present in cells from A_2A_KO mice, suggesting a compensatory adaptation generated by a lack of A_2A_. The underlying mechanism of the potential crosstalk between A_2A_ and A_3_ and how it may affect endothelial function and angiogenesis are unknown. We reported previously that cell migration is mainly related to the activation of A_2A_ and A_3_, but not A_2B_ receptors in a primary culture of human endothelial progenitor cells [31]. Therefore, we encourage future studies to try to understand the potential A_3_-mediated pro-angiogenic behavior of endothelial cells that lack A_2A_ receptors.

However, our study has some limitations due to the combined information gathered from in vitro and in vivo experiments. For the former, we used a primary culture of pulmonary endothelial cells, while in the latter, we used a skin wound healing model. However, our CD31-enriched primary culture presented functional angiogenic capacities, which indeed were the focus of our research. Despite that, we cannot rule out the possibility of functional changes in the interaction between A_2A_ and ER in other types of endothelial cells, including those from the skin microcirculation. One intriguing result was the finding of sex dimorphism in blood perfusion in the wounded area, in which male WT mice had higher perfusion than female WT mice. As indicated previously, this difference may reflect the degree of healing, but also might indicate a limitation in the laser penetration of the Doppler analysis.

In conclusion, our results indicate that female mice exhibited advantages in the wound healing process which is associated with 17β-estradiol upregulation of A_2A_-mediated angiogenesis in a primary culture of female endothelial cells. The potential underlying mechanism for this effect may involve translational rather than transcriptional regulation of the A_2A_ receptor through activation of ERα and ERβ receptors, although regulatory feedback between ER and A_2A_ expression might be also present. The interaction between ER and A_2A_ in the regulation of angiogenesis brings a new area of research into the complex regulatory scenario of the healing process.

## 4. Materials and Methods

### 4.1. Reagents

The A_2A_ adenosine receptor selective agonist, 2-*p*-(2-Carboxyethyl) phenethylamino-5′-N-ethylcarboxamido adenosine hydrochloride hydrate (CGS-21680), and the non-selective agonist, 5′-(N-ethylcarboxamido) adenosine (NECA), as well as the A_2A_ adenosine receptor selective antagonist, 4-(2-(7-Amino-2-(2-furyl) (1,2,4) triazolo (2,3-α) (1,3,5)triazin-5-ylamino]ethyl) phenol (ZM-241385), were purchased from Tocris Biosciences (Bristol, UK). The non-selective estrogen receptor agonist, 1,3,5-estratriene-3,17β-diol (17β-estradiol), was from Sigma-Aldrich, (San Luis, MO, USA). Antagonist selective estrogen receptor alpha (ERα), 1,3-*Bis*(4-hydroxyphenyl)-4-methyl-5-(4-(2-piperidinylethoxy) phenol)-1*H*-pyrazole dihydrochloride (MPP), and beta (ERβ), 4-(2-Phenyl-5,7-*bis*(trifluoromethyl) pyrazolo(1,5-*a*) pyrimidin-3-l) phenol (PHTPP), were also purchased from Tocris Biosciences, UK.

### 4.2. Animals

C57BlackL/6 mice were purchased from the animal facility of the Pontificia Universidad Católica de Chile (PUC). Dr. Jiang-Fan Chen from Boston University, USA, donated A_2A_KO mice. The generation of A_2A_KO mice has been described in detail previously [32]. In brief, an A_2A_ receptor genomic fragment was split by a positive selection marker (neomycin cassette) which replaced the 3′ end of exon 2, the adjacent 5′ splice junction, and intron sequences. Confirmation of A_2A_KO was performed using the amplification of neomycin cassette using PCR (Appendix A). Mice were housed at the Universidad de Valparaiso, Chile animal facility where they were kept under standard environmental conditions which included controlled temperature (25 °C) and humidity, exposure to 12/12 h light/darkness cycles, and food and water supply ad libitum. All experiments were performed independently of estrous cycle in the case of female mice. This study was carried out following the recommendations of the guidelines for the Care and Use of Laboratory Animals published by the US National Institute of Health. The Ethical Committee from the Universidad del Bio Bío (UBB) and FONDECYT (1140586, Chile) approved the protocol (1 March 2014).

### 4.3. In Vivo Wound Healing Assay

The in vivo wound healing assay was performed as described previously by our laboratory [33]. Briefly, female and male WT and A_2A_KO mice (four animals per group, three months old; body weight 20–23 g) were isolated in individual cages. Animals were anesthetized using ketamine (100 mg/kg) under controlled temperature (37 °C) and aseptic conditions. After that, the animals were shaved at the dorsal level to make a wound (5 mm) using a punch. An immediate-bonding adhesive was used to isolate the injured area. A follow-up analysis was performed on day 10 and photographs were taken every two days to record the evolution of wound healing. Percentage of wound closure was calculated as follows: wound healing = (*A*_0_ − *A_n_*/*A*_0_).

Where wound healing represented wound closure. *A*_0_ represented the wound area at time 0, and *A_n_* represented the wound area at “*n*” days follow-up.

### 4.4. Isolation of Mice Lung Endothelial Cells (mPEC)

Mice pulmonary endothelial cells (mPEC) were isolated following a similar protocol that was used for human placental microvascular endothelial cells (hPMEC) [34]. Briefly, the lungs were excised, immersed in Medium 199 (M119) (Life Technologies, Carlsbad, CA, USA), and kept on ice. Lung tissue was cut into pieces of approximately 1 mm^3^, passed through a surgical mesh, and centrifuged at 250× *g* for 10 min. Samples were digested for 2 h at 37 °C, using 0.1 mg/mL of collagenase type II (Life Technologies). After enzymatic digestion, samples were washed using phosphate buffer solution (PBS) + 0.1% fetal bovine serum (FBS) and centrifuged three times at 250× *g* for 10 min each. Digested tissue was resuspended in M199, which contained 5 mM D-glucose, 20% new-born calf serum (NBCS), 20% fetal calf serum (FCS), 3.2 mM L-glutamine, and 100 U/mL penicillin-streptomycin (primary culture medium, PCM). The cell suspension was transferred to 1% gelatin-coated T_25_ culture flasks for culture (37°C, 5% O_2_, 5% CO_2_) in PCM until confluence. Confluent cells were trypsinized (trypsin/EDTA = 0.25/0.2%) (Life Technologies, Carlsbad, CA, USA) at 3 min, 37 °C and subjected to CD31 positive immune-selection using Dynabeads^©^ CD31 (Thermo Fisher, Waltham, MA, USA) [34]. Immune-selection of endothelial cells was performed by mixing (20 min, 4 °C) magnetic microbeads conjugated to *anti*-CD31 antibody (platelet endothelial cell adhesion molecule 1, PECAM-1 or CD31) with the cell suspension to yield 48 × 10^3^ beads/μL of cell suspension. Cells attached to the magnetic microbeads were collected and washed (3×) in PBS at 37 °C). CD31-coated microbead-attached cells were re-suspended in PCM containing 10% NBCS and 10% FCS and cultured until passage 2 [35]. No analysis of the effect of passages was performed; inter-assay variation in the proliferation assays was 18.3 ± 1.2%.

Endothelial cells were further characterized by Western blot analysis for endothelial markers (hematopoietic progenitor cell antigen CD34 and vascular endothelial growth factor receptor type 2 or KDR, see below) and by in vitro angiogenesis assay (tube formation on Matrigel). Briefly, pulmonary endothelial cells (4 × 10^4^) from WT or A_2A_KO mice were cultured on a 96-well plate coated with 40 μL Matrigel basement membrane matrix (Merck, Darmstadt, Germany). Assays were performed at different serum concentrations (0.1–1%). The formation of branches was quantified using the “Angiogenesis Analyzer” plugin from ImageJ V1.48 software.

### 4.5. Semiquantitative and Quantitative PCR

Total RNA was isolated using Trizol^®^ Reagent (Life Technology, Carlsbad, CA, USA) according to the manufacturer’s instructions. RNA concentration was measured using MaestroNano (Maestrogen, Xiangsham, Hsinchy, Taiwan). cDNA was synthesized from 1 μg of RNA total. The reverse transcription was performed as previously described [31] using a high capacity cDNA RT kit (Life Technology) according to the manufacturer’s instructions.

Details of primers used in quantitative PCR (Q-PCR) are described in Appendix A and comprised A_1_, A_2B_, and A_3_ as well as *mlp37* as housekeeping. The PCR reaction was performed in a final volume of 20 μL which included 1 μL of cDNA, 200 nM primers, 10 μL Brilliant II SYBR Green Q-PCR master mix (including SureStart Taq DNA polymerase), and 0.375 μL reference dye (5 μM) (Agilent Technologies, Santa Clara, CA, USA). Amplification was performed in a Rotor Gene 6000 thermocycler (Corbett Life Science, Brisbane, Australia). Q-PCR cycles were set up as follows: 35 cycles of denaturation (95 °C, 30 s), annealing (see Appendix A for melting temperature of individual genes, 60 s), and extension cycles (72 °C, 60 s) with a final extension at 72 °C (5 min). Fluorescent products were detected in the third step of cycling. Product specificity was confirmed by agarose gel electrophoresis (2% *v*/*v*) and melting curve analysis. Quantification of gene expression was performed following the delta-delta CT method [36].

For semiquantitative PCR, the mRNA levels of *Adora2a* (adenosine receptor A_2A_), *Jarid1c/1d*, and *mlp37* genes were assessed using a commercially available kit (Multigene Gradient, Labnet, Edison, NJ, USA). Cells were sexed by specific amplification of the *Jarid1c* and *Jarid1d* gene, which yields double bands for males and a single band for females, as previously reported (Clapcote and Roder, 2005). For PCR analysis, 20 µL reactions were carried out using DreamTaq Green PCR Master Mix 2X (Thermo Scientific, Waltham, MA, USA) with 1 µM primers according to the manufacturer’s instructions. PCR products were separated using electrophoresis in 1.5% agarose gels and visualized using ethidium bromide under UV light. The primers used are included in Appendix A. DNA polymerase was activated at 95 °C (10 min) followed by 35 cycles at 95 °C (30 s), 57 °C (60 s), and 72 °C (60 s) with a final extension at 72 °C (5 min). Fluorescent products were detected in the third step of cycling. Product specificity was confirmed by agarose gel electrophoresis (1.5% *v*/*v*). For each sample, the target genes were normalized to that of the housekeeping gene *mlp37*.

### 4.6. Western Blot

Cellular proteins were extracted using lysis buffer (Tris HCL, pH 8, 20 mM; NaCl 137 mM; EDTA 2 mM; glycerol 10%; Nonidet P-40 1%) that contained a protease inhibitor cocktail (Thermo Scientific, Waltham, MA, USA). Cell extracts were centrifuged at 14,000× *g* for 10 min at 4 °C. Proteins (70 μg) from the supernatant were separated using SDS-PAGE (10%), transferred to nitrocellulose membranes, and probed with primary antibodies: CD34 (Abcam, Cambridge, UK; ab8158, dilution 1:2500 *v*/*v*); vascular endothelial growth factor receptor 2 (KDR) (Cell Signalling Technology, Danvers, MA, USA; #2479, dilution 1:1000 *v*/*v*); A_2A_ receptor (Millipore, Burlington, MA, USA; dilution 1:2000 *v*/*v*); ERα (Santa Cruz, Dallas, TX, USA; # sc-8002, dilution 1:1000 *v*/*v*); ERβ (Santa Cruz, # sc-390243, dilution 1:1000 *v*/*v*); VEGF (Abcam, MA, USA, #9479, 1:1000 *v*/*v*); and β-actin (Sigma-Aldrich; St Louis, MO, USA; clone AC-74, dilution 1:15000 *v*/*v*). Rabbit (Thermo Scientific) or mouse (Sigma-Aldrich) secondary antibody conjugated with horseradish peroxidase were used for visualization. Bands on gels were scanned and images quantified using ImageJ V1.48 software (National Institute of Health, USA) as previously described [37].

### 4.7. Cell Proliferation

Pulmonary endothelial cells (7 × 10^3^ cell/mL) from WT or A_2A_KO mice were seeded in 96-well plates and maintained in standard growth conditions (PCM 37°C, 5% CO_2_). After serum deprivation, cells were incubated with 5-bromo-2-deoxyuridine (BrdU) in the presence (24 h) of adenosine deaminase (ADA, 0.1 U/mL) either alone or in combination with the following agonists: 17β-estradiol (ER, non-selective agonist, E2; 10^−7^ M), CGS-21680 (A_2A_, selective agonist, 10^−5^ M), and NECA (adenosine receptor non-selective agonist, 10^−5^ M), or with the selective antagonists for ERα (MPP, 10^−6^ M) and ERβ (PTHPP, 10^−6^ M) as well as for the A_2A_ adenosine receptors (ZM-241385, 10^−4^ M) in parallel experiments. Cells were fixed and measurements were performed according to the manufacturer’s instructions (Roche Diagnostics, IN, USA), as we previously reported [20]. 

### 4.8. Cell Migration

Cell migration was analyzed in vitro as described [37,38]. Briefly, cells were allowed to reach confluence in growth medium, and switched to serum-free medium. The monolayer was wounded with a single sterile cell scraper of constant diameter (0.2 cm). After migration, cells were fixed using absolute ethanol (200 proof) for 20 min, washed three times with PBS, and stained using hematoxylin. Cells were observed at 40× magnification on a phase contrast inverted microscope (Olympus, Tokyo, Japan). Six random images were taken using a digital camera (MShot MD90, Guangzhou Micro-shot Technology Co., Ltd., Guangzhou, China) immediately after wound generation and 24 h after treatment with adenosine A_2A_ receptor agonist CGS-21680 (10^−5^ M). Cell migration was analyzed using the area measurement plugin from ImageJ software. Migratory area was expressed as percentage of migration into the denudated area
*M. area* = (*A*_0_ − *A*_24h_/*A*_0_) ∗ 100
where *M. area* represented the migratory area, *A*_0_ represented the area at time 0, or denudated area, and *A*_24h_ represented the area that remained denude after 24 h.

### 4.9. In Vitro Angiogenesis

Pulmonary endothelial cells (4 × 10^4^) from wild type or A_2A_KO mice were cultured on a 96-well plate pre-coated with 40 μL Matrigel basement membrane matrix (Corning Labware, MA, USA). Assays were performed in absence or presence (6 h) of adenosine A_2A_ receptor agonist, CGS-21680 (10^−5^ M). Tubes were photographed using inverted phase contrast microscope under 10× magnification (Olympus, Tokyo, Japan). Formation of networks (branches) was quantified using the “Angiogenesis Analyzer” plugin from ImageJ 1.48 software as previously reported [37,38].

### 4.10. Statistical Analysis

The variables were analyzed using non-parametric ANOVA tests. Mann–Whitney tests were used for pair-comparisons in cases where significant differences (*p* < 0.05) were found. Values are presented as media ± S.E.M., and *p* < 0.05 were considered statistically significant. GraphPad Prism V5.00 (GraphPad Software, Inc., San Diego, CA, USA) was used for data and statistical analysis.

## 5. Conclusions

Female mice exhibited advantages in the wound healing process which is associated with 17β-estradiol upregulation of A_2A_-mediated angiogenesis in a primary culture of female endothelial cells. The potential underlying mechanism for this effect may involve translational rather than transcriptional regulation of the A_2A_ receptor through activation of ERα and ERβ receptors, although regulatory feedback between ER and A_2A_ expression might be also present.

## Figures and Tables

**Figure 1 ijms-21-07145-f001:**
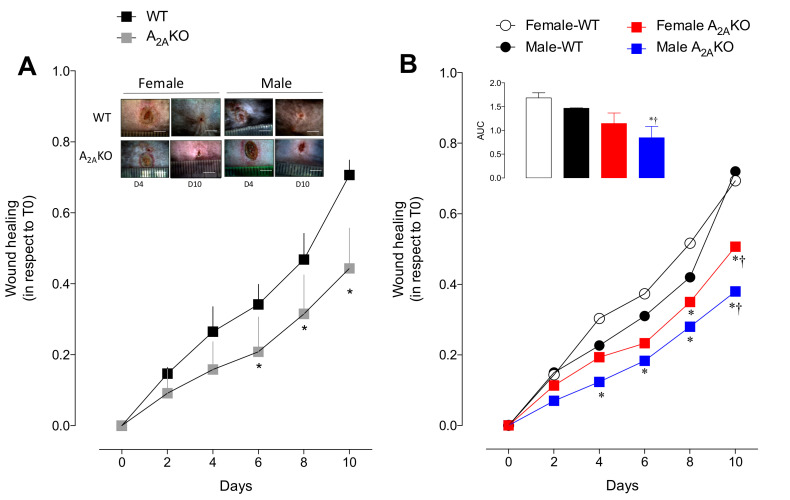
Wound healing in vivo assay in wild type and A_2A_KO mice: (**A**) Time course (up to 10 days) of wound healing (relative wound closure with respect to day 0) in WT and A_2A_KO mice. Insert includes representative images of the wounded area at day 4 (D4) and day 10 (D10) after injury. The line in the representative images represents 0.5 cm. (**B**) Effect of sex on the wound healing process studied as in A. Inset represents area under the curve (AUC) of the healing process in female and male WT and A_2A_KO mice. In A, * *p* < 0.05 versus corresponding value in WT. In B, * *p* < 0.05 versus female WT mice. † *p* < 0.002 versus male WT. Values were expressed as mean ± SEM and *n* = 3–4 per group.

**Figure 2 ijms-21-07145-f002:**
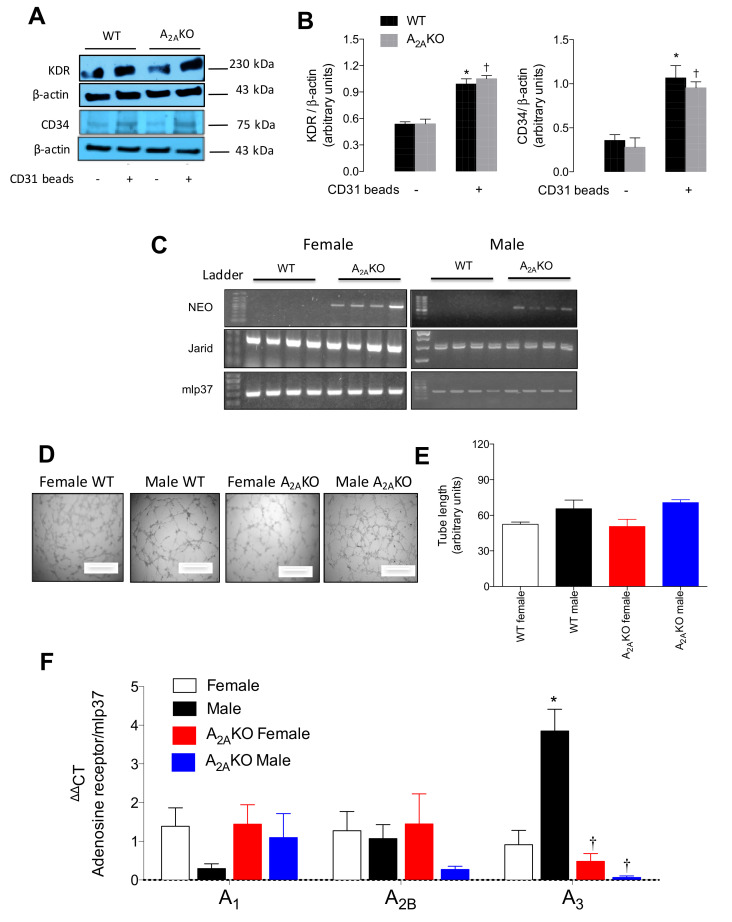
Expression of adenosine receptors in female and male pulmonary endothelial cells (mPEC) isolated from WT and A_2A_KO mice. Pulmonary endothelial cells were isolated using collagenase-mediated tissue digestion and immune-selection with CD31-coated Dynabeads. (**A**) Representative blots of endothelial cell markers. VEGF receptor 2 (KDR, ~230 kDa), CD34 (~80 kDa) and β-actin (~43 kDa) were identified in cell extraction of immunoselected (CD31^+^, Waltham, MA, USA) cells derived from wild type (WT) and A_2A_-deficient mice (A_2A_KO). Cells that were immunoselected are identified with a plus sign (+). (**B**) Semiquantitative densitometry of KDR/β-actin and CD34/β-actin ratio. (**C**) Confirmation of gender and genetic background of mPEC isolated from females (single band in the PCR for *Jarid* gene) or male mice (double band). A_2A_KO cells were identified by positive amplification of neomycin cassette (NEO). *mlp37* gene was used as housekeeping. DNA Ladder 100 bp. (**D**) In vitro angiogenesis assay at 4 h of incubation with bovine serum (1%). (**E**) Quantification of tube length of angiogenesis in vitro. (**F**) QPCR analysis of mRNA levels of A_1_, A_2B_, and A_3_ adenosine receptors in male and female WT and A_2A_KO mice. See Appendix A for details about primers and PCR amplicons. In (**B**) * *p* < 0.05 versus CD31^-^ cells in WT mice. † *p* < 0.05 versus CD31^-^ cells in A_2A_KO. In (**F**). * *p* < 0.05 versus respective value in female WT mice. † *p* < 0.05 versus respective value in male WT mice. Values were expressed as mean ± SEM. *n* = 3–5 per group. All experiments were performed in duplicate.

**Figure 3 ijms-21-07145-f003:**
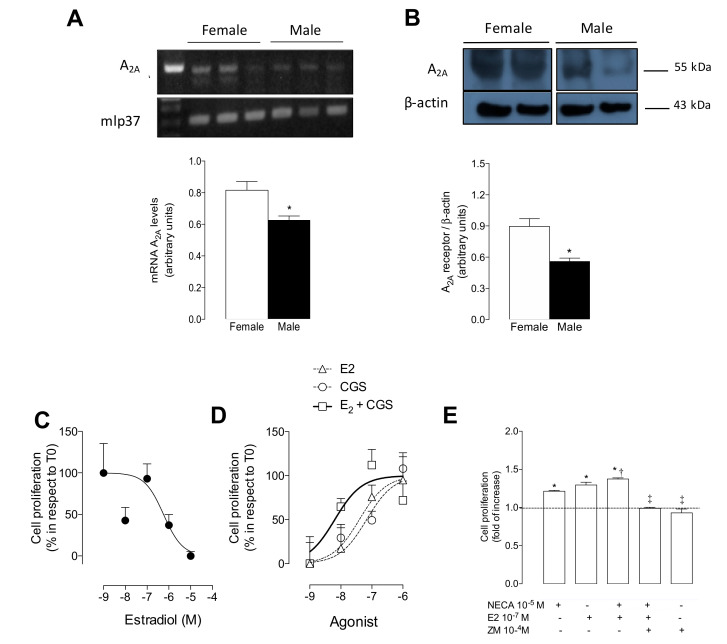
Expression of A_2A_ and cell proliferation in mPEC derived from female and male wild type mice. (**A**) PCR analysis of mRNA levels of A_2A_ and housekeeping gene *mlp37*. Bars below represent densitometry of A_2A_/*mlp37* ratio in female (white bars) and male (black bars) mPEC from wild type (WT) mice. (**B**) Western blot of A_2A_ (~55 kDa) and β-actin (~43 kDa) proteins. Bars below represent semiquantitative densitometry of A_2A_/β-actin ratio. Space between blots means that they were run in different gels. (**C**) Dose–response curve of cell proliferation stimulated by 17β-estradiol (10^−9^ to 10^−5^ M, for 24 h) in mPEC isolated from male WT mice. (**D**) Dose–response curve of cell proliferation stimulated by 17β-estradiol (triangles) or CGS-21680 (circles) or both (squares) in mPEC isolated from female WT mice. (**E**) Cell proliferation in the presence of NECA (10^−5^ M, 24 h), with (+) or without (-) 17β-estradiol (10^−7^ M) or ZM-241385 (10^−4^ M). In (**A**) and (**B**), * *p* < 0.05 versus respective value in female WT mice. In (**E**), * *p* < 0.05 versus basal condition (i.e., without agonist, represented by scatter line) in mPEC isolated from female WT mice. † *p* < 0.01 versus NECA. ‡ *p* < 0.0001 versus NECA + 17β-estradiol. Values were expressed as mean ± SEM, *n* = 5–11 per group. All experiments were performed in duplicate.

**Figure 4 ijms-21-07145-f004:**
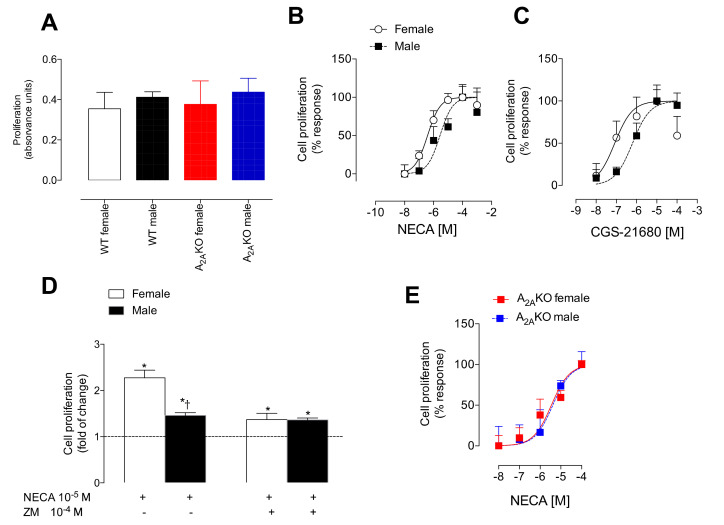
Sex dimorphism in the A_2A_-mediated cell proliferation. (**A**) Cell proliferation measured by bromouridine incorporation at basal conditions in female and male WT and A_2A_KO mice. Respective groups are identified by different colors. (**B**) Dose–response curves of cell proliferation induced by NECA or (**C**) CGS-21680. (**D**) Cell proliferation in female (white bars) and male (black bars) WT mice with (+) NECA (10^−5^ M × 24 h) or ZM-241385 (10^−4^ M × 24 h). The dotted line represents basal values (i.e., control without agonists). (**E**) Dose–response curves of cell proliferation induced by CGS-21680 in female and male A_2A_KO. In (**D**), * *p* < 0.05 with respect to basal, † *p* < 0.05 with respect to female WT. Values were expressed as mean ± SEM, *n* = 4–8 per group in duplicates.

**Figure 5 ijms-21-07145-f005:**
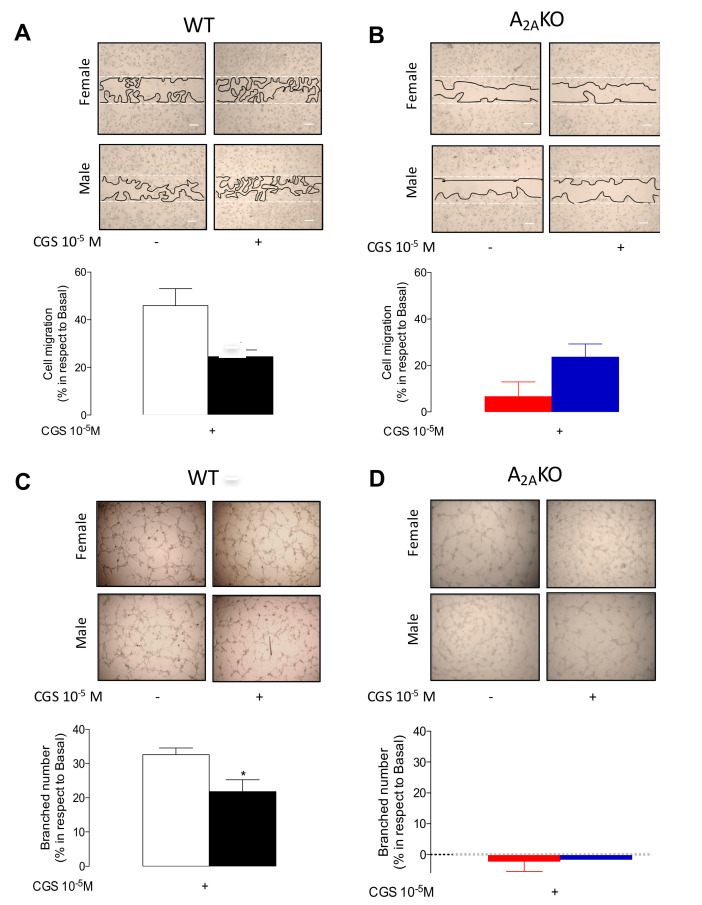
Cell migration and in vitro angiogenesis in mPEC isolated from female and male WT and A_2A_KO mice. Cell migration was analyzed using the in vitro wound closure assay. (**A**) Representative images of migration assays in mPEC derived from female and male WT mice. Bars represent the percentage of increase in cell migration in the presence (+) of CGS-21680 (10^−5^ M × 24 h) in WT female (white bars) and male WT mice (black bars). (**B**) Representative images of migration assay in mPEC derived from female and male A_2A_KO mice. Bar represents, as in (**A**), the response of mPEC isolated from female (red bars) and male (blue bars) A_2A_KO mice. (**C**) Representative images and respective quantification of branched number in the in vitro angiogenesis assay (see Methods) in cells treated as in (**A**) in WT or (**D**) A_2A_KO mice. * *p* < 0.05 vs. female WT. Values are mean ± SEM, *n* = 4–6 per group in duplicates.

**Figure 6 ijms-21-07145-f006:**
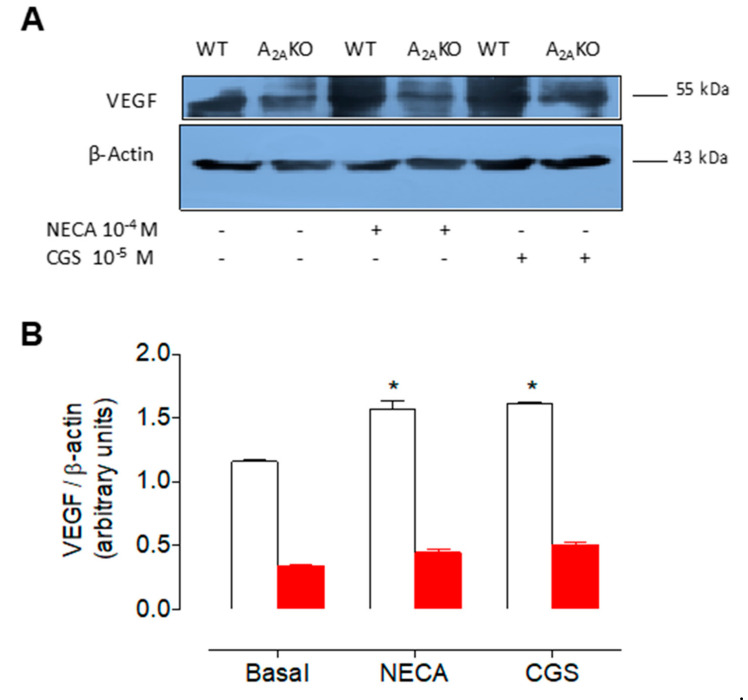
Expression of VEGF in female mPEC derived from wild type and A_2A_KO mice. (**A**) Representative Western blot of VEGF (~55 kDa) and β-actin (~43 kDa) in mPEC from female WT (white bars) and A_2A_KO mice (red bars). Cells were incubated (12 h) with (+) or without (-) NECA (10^−4^ M) or CGS-21680 (10^−5^ M). (**B**) Semiquantitative densitometry of VEGF/β-actin ratio as in (**A**). * *p* < 0.05 respect to basal. Values are mean ± SEM, *n* = 4 per group in duplicates.

**Figure 7 ijms-21-07145-f007:**
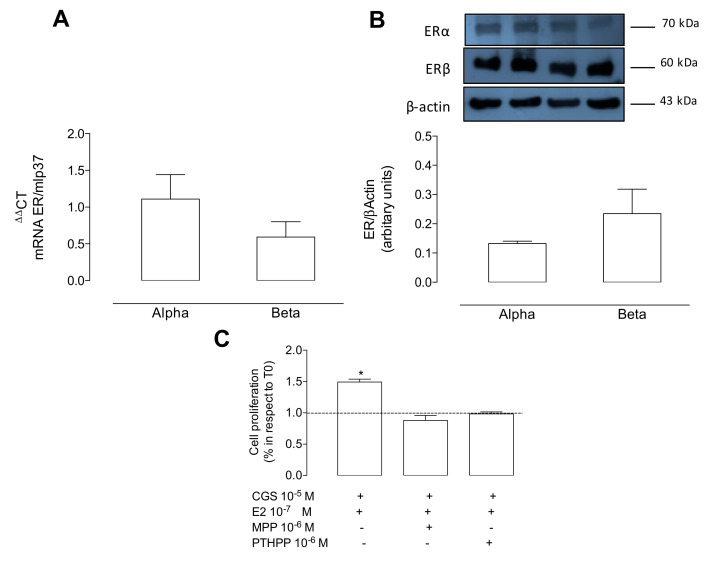
17β-estradiol potentiates A_2A_-mediated cell proliferation via estrogen receptors in female mPEC. (**A**) Q-PCR for measuring mRNA of estrogen receptor (ER) alpha and beta. Both genes were normalized to *mlp37*. *p* = 0.34 between alpha and beta. (**B**) Representative Western blot and respective densitometry of ERα (~70 kDa), ERβ (~60 kDa), and β-actin (~43 kDa). ERs/β-actin ratio is presented. *P* = 0.86 between alpha and beta. (**C**) Cells incubated with 17β-estradiol + CGS-21680 with or without the antagonists MMP (10^−6^ M × 24 h) or PHTPP (10^−6^ M × 24 h) for ERα and ERβ, respectively. Dotted line in C represents basal values (i.e., control without stimuli). * *p* < 0.05 versus control. Values are mean ± SEM, *n* = 3–6 per group. All experiments were performed in duplicates.

**Table 1 ijms-21-07145-t001:** LogEC_50_ of 17β-estradiol and CGS-21680 for female mPEC proliferation in WT mice.

Analyzed Parameters	17β-Estradiol	CGS-21680	17β-Estradiol + CGS-21680
LogEC_50_	−7.40	−7.17	−8.19 *
95% confidence intervals	−8.24 to −6.57	−7.67 to −6.67	−8.92 to −7.47
Number of point analyzed	58	58	60
Outliers (excluded, Q = 1.0%)	0	0	0

* *p* < 0.05 versus 17β-estradiol concentration. Values are mean ± SEM. All experiments were performed in triplicate.

**Table 2 ijms-21-07145-t002:** Log EC_50_ for NECA and CGS-21680 in mPEC isolated from WT and A_2A_KO mice.

Agonists	Wild type		A_2A_KO	
	Female	Male	Female	Male
NECA	−6.4 (−6.7 to −6.0)	−5.5 (−6.1 to −4.9) *	−5.4 (−5.9 to −5.0) *	−5.3 (−6.1 to −4.6) *
CGS-21680	−7.1 (−7.6 to −6.5)	−6.2 (−6.5 to −5.8) *	--	--

In parentheses, 95% confidence interval (CI). All experiments were performed in triplicate. * *p* < 0.05 versus female WT.

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
