# Peer review of "Advantages in Wound Healing Process in Female Mice Require Upregulation A_2A_-Mediated Angiogenesis under the Stimulation of 17β-Estradiol"

_ijms, 2020, doi:10.3390/ijms21197145_

Round 1
Reviewer 1 Report
Comments to ijms-895458
Advantages in wound healing process in female mice require up-regulation A2A–
mediated angiogenesis under the stimulation of 17β- estradiol
Estrogenic steroids and adenosine A2A receptors promote the wound-healing and the angiogenesis processes. However, so far it is unclear whether estrogen may regulate the expression and pro-angiogenic activity of A2A receptors. Therefore, the authors would like to evaluate whether estrogen regulates the expression and/or pro-angiogenic activity of A2A adenosine receptors likely involves the activation of either ERα or ERβ receptors or both estrogen receptors. Sexual dimorphism in wound healing observed in the A2AKO mice process reinforces the functional crosstalk between ER and A2A receptors.
Comment:
- The quality of Fig 2A and 2N needs further improvement, since the control band by β-actin bands seemed to hard convince the audience for their presentation. To compare the band of KDR between A2AKO and WT, it seemed that expression pattern of KDR was weak in the A2AKO mice, and by contrast, in the results of Fig 2B, the expression pattern seemed to be higher in the A2AKO bar?
- The same critiques were also present in the expression of CD 34, and pattern seemed to be discrepancy between Figure 2 A and Figure 2B.
- There was absent of description of the methods and materials for the results of the Fig5 about in vitro wound closure assay and in vitro angiogenesis assay.
- The discrepancy of the results presentation was found in the Figs 6Aand 6B. I think that internal control of beta-actin might need validation. The expression of VEGF and beta-actin seemed to be lower in the A2AKO, but after qualification of bar figure, the results seemed to be reverse.
- The discrepancy of gene expression and protein expression was found in ER alpha and ER beta, and these data were shown in the Fig7A and Fig7B focusing the study for mPEC cells in the wild-type female mice.
- There is absence of some methods to evaluate the primers in the Table S1, although the authors have shown that should be present in the line 436 and 437.
- In the line 448, the authors would like to study Jarid1c and Jarid1d, and would like to shown the primers in both genes, but they were absent in the Table S1, and of most importance, these data after evaluation and study did not appear in the current article.
- In general, the current article is well done and worthy of consideration.
Author Response
We greatly appreciate all these valuable comments because they have corrected our mistakes and improved the current version of our manuscript.
Here are included our replies to each one of her/his observations:
RE 1, 2: Thanks for these observations. We have included a more representative image of the western blot analysis of both KDR and CD34. Also, we have performed new densitometry in all western blots. These measurements did not change the original findings but represent more accurately what is reported. Please see lines 119-123.
RE 3: Apologies for this omission. We have incorporated the description of both methods. Please, check line 497-517.
RE 4: Apologies because lack of the identification in each bar leads to this misinterpretation. We have incorporated that identification, which agrees with this reviewer's comment regarding the lower expression of VEGF in A2AKO mice. Also, a new blot of beta-actin with longer exposition is incorporated.
RE5: We respectfully disagree with this comment. Because, differences in mRNA (n=4, p=0.34) and protein levels (n=6, p=0.86) of ER alpha and ER beta did not reach statistical significance. Then, according to this analysis, the expression of both receptors are similar in mPEC cells isolated from wild type female mice. To clarify that, we have added the respective p-value in the figure legend.
RE6: Apologies for this omission. We have incorporated the respective melting temperature in Table S1.
RE7: We have edited the identification of Jarid gene primers. To clarify, those primers allow identification of both Jarid1c and Jarid1d, which in turn yields double bands for males and a single band for females. Regarding the inclusion of this gene in the entire manuscript, we must indicate that because we are interested in sex-mediated differences, cell cultures are demandingly identified through the whole process (since isolation to final experimentation) in our protocols. To confirm that, we have included this sex-identification in Figure 2.
RE8: thanks for all these valuable comments.
Reviewer 2 Report
The manuscript entitled “Advantages in wound healing process in female mice require up-regulation A2A–mediated angiogenesis under the stimulation of 17β- estradiol” focuses on the regulation of estrogen on the expression and pro-angiogenic activity of A2A receptors.
The manuscript is well written, it has a complete experiment design, and the discussion is detailed and correct. The results obtained are clear. I have some comments:
-The authors should improve Figure 2 A indications, they are not so clear to the reader.
The resolution of β-actin (Figure 6A) is not ideal. Please, provide a new western blot analysis.
-The authors performed experiments in female and male mice. Did you measured estrogen levels in A2A KO mice?
-Did you perform estrous cycle monitoring?
Author Response
We appreciate all these valuable comments. Also, we acknowledge some limitations in our study based on those comments.
Regarding specific replies to issues raised by this review, here are incorporated.
RE1: We have incorporated additional description in the figure legend.
RE2: We have incorporated a new blot with a longer exposition.
RE3: Thanks for raising this important point. But, unfortunately, we did not measure estrogen levels.
RE4: Thanks for this comment. But, unfortunately, we did not monitor the estrous cycle in our experimental setting. We have declared this limitation in the method section. Line 400-401.